# Predictive Chemistry Augmented with Text Retrieval

**Yujie Qian**[†‡]  **Zhening Li**[‡]  **Zhengkai Tu**[‡]  **Connor W. Coley**[‡§]  **Regina Barzilay**[†‡]

[†]Computer Science and Artificial Intelligence Lab, MIT
[‡]Department of Electrical Engineering and Computer Science, MIT
[§]Department of Chemical Engineering, MIT
{yujieq,regina}@csail.mit.edu   {zli11010,ztu,ccoley}@mit.edu

## Abstract

This paper focuses on using natural language descriptions to enhance predictive models in the chemistry field. Conventionally, chemoinformatics models are trained with extensive structured data manually extracted from the literature. In this paper, we introduce TextReact, a novel method that directly augments predictive chemistry with texts retrieved from the literature. TextReact retrieves text descriptions relevant for a given chemical reaction, and then aligns them with the molecular representation of the reaction. This alignment is enhanced via an auxiliary masked LM objective incorporated in the predictor training. We empirically validate the framework on two chemistry tasks: reaction condition recommendation and one-step retrosynthesis. By leveraging text retrieval, TextReact significantly outperforms state-of-the-art chemoinformatics models trained solely on molecular data.

## 1 Introduction

In this paper, we propose a method for leveraging automatically retrieved textual knowledge to improve the predictive capacity of chemistry models. These chemoinformatics models are utilized for a wide range of tasks, from analyzing properties of individual molecules to capturing their interactions in chemical reactions (Yang et al., 2019; Segler et al., 2018; Coley et al., 2018). However, standard approaches make these predictions operating solely on molecular encodings. Despite significant advances in neural molecular representations, the accuracy of these models still offers room for improvement. We hypothesize that their performance can be further enhanced using relevant information from the scientific literature.

For instance, consider the task of finding a catalyst for the chemical reaction shown in Figure 1. This prediction proves to be challenging when considering the reaction components alone (Gao et al.,

2021). At the same time, scientific literature provides several cues about potential catalysts (see highlighted excerpts). While on their own these paragraphs might not provide a comprehensive answer, they can guide a molecular predictor. This intuition motivates our approach to aggregating textual sources with molecular representations when reasoning about chemistry tasks. Specifically, we aim to align the representations of the reaction and its corresponding text description, enabling the model to operate in the combined space. This design not only enables the model to retrieve readily available information from the literature but also enhances its generalization capacity for new chemical contexts.

We propose TextReact, a novel predictive chemistry framework augmented with text retrieval. TextReact comprises two modules – a SMILES-to-text retriever[1] that maps an input reaction to corresponding text descriptions, and a text-augmented predictor that fuses the input reaction with the retrieved texts. The model learns the relation between a chemical reaction and text via an auxiliary masked LM objective incorporated in the predictor training. Furthermore, to improve generalization to unseen reaction classes, we simulate novel inputs by eliminating from the training data the closest textual descriptions for given reactions.

In addition to condition recommendation, TextReact can be readily applied to other chemistry tasks. In our experiments, we also consider one-step retrosynthesis (Coley et al., 2017), the task of predicting reactants used to synthesize a target molecule (see Figure 2). By leveraging relevant text, TextReact achieves substantial performance improvement compared to the state-of-the-art chemoinformatics models trained on reaction data alone. For instance, for condition recommen-

---

[1]SMILES, i.e. Simplified Molecular-Input Line-Entry System, is a specification in the form of a line notation for describing chemical structures (Weininger, 1988).

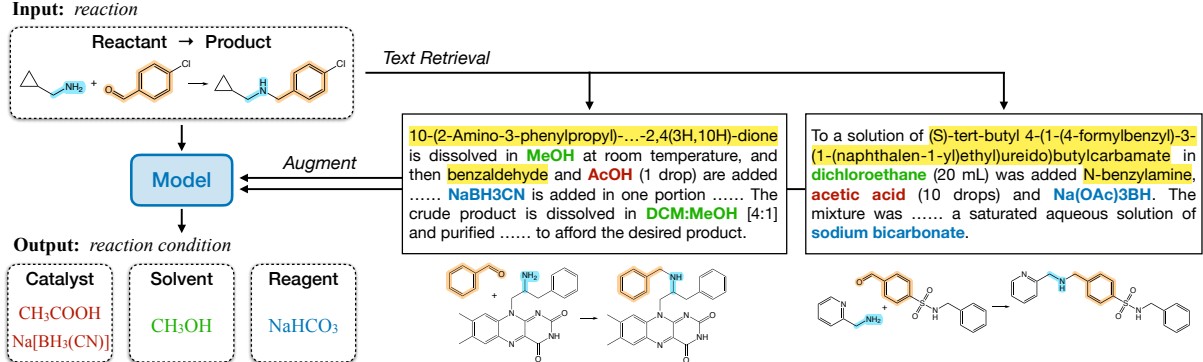

Figure 1: Predictive chemistry augmented with text retrieval. For example, given the task of reaction condition recommendation, we retrieve texts relevant to the input reaction to provide additional evidence for the model's prediction. The two retrieved texts describe similar reactions to the input and share similar conditions. For visualization purposes, we mark the catalyst, solvent, and reagent in red, green, and blue in the retrieved texts. However, we do not assume that the text corpus contains such structured data.

dation, TextReact increases the top-1 prediction accuracy by 58.4%, while the improvement in one-step retrosynthesis is 13.6–15.7%. The improvement is consistent under both random and time-based splits of the datasets, validating the efficacy of our retrieval augmentation approach in generalizing to new task instances. Our code and data are publicly available at `https://github.com/thomas0809/textreact`.

## 2 Related Work

**Multimodal Retrieval** This field studies retrieval algorithms when the query and target are in different modalities, as exemplified by image-text retrieval (Weston et al., 2010; Socher and Fei-Fei, 2010; Socher et al., 2014; Karpathy et al., 2014; Faghri et al., 2018). Previous research learns multimodal embeddings in image and text using techniques ranging from kernel methods to more expressive neural networks. This line also extends to other modalities such as video (Miech et al., 2020) and audio (Aytar et al., 2017).

Our retrieval method closely relates to CLIP (Radford et al., 2021), a multimodal model that leverages contrastive learning to align images with their corresponding natural language descriptions. Moving to the chemistry domain, Edwards et al. (2021) proposed Text2Mol to retrieve molecules using natural language queries. Our SMILES-to-text retriever operates in the opposite direction, i.e., we use the aligned embedding spaces to retrieve a text description given a reaction query. We then use the retrieved text to enhance chemistry prediction.

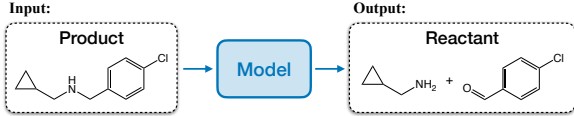

Figure 2: One-step retrosynthesis, another predictive chemistry task studied in this paper.

**Retrieval-Augmented NLP** There have been several studies on augmenting NLP models with retrieval. The idea is to retrieve relevant documents from a corpus and use them as additional context to the model. Earlier works on open-domain question answering adopted standalone retrievers to identify supporting paragraphs for the given questions (Chen et al., 2017; Lee et al., 2019; Karpukhin et al., 2020). Guu et al. (2020) jointly trained an end-to-end retriever with a language model, and Lewis et al. (2020b) extended the idea to general sequence-to-sequence generation.

Our work implements a similar idea of retrieval augmentation. However, we focus on tasks in the chemistry domain (see Figures 1 and 2), with the goal of enhancing chemistry prediction models with natural language descriptions retrieved from the literature.

**Natural Language Grounding** Previous studies have demonstrated how natural language can be harnessed as a grounding mechanism to supervise tasks in various domains. For example, in computer vision, natural language descriptions have been used to improve fine-grained classification of bird images (He and Peng, 2017; Liang et al., 2020) and to regularize the learning of visual representations (Andreas et al., 2018; Mu et al., 2020). In rein-

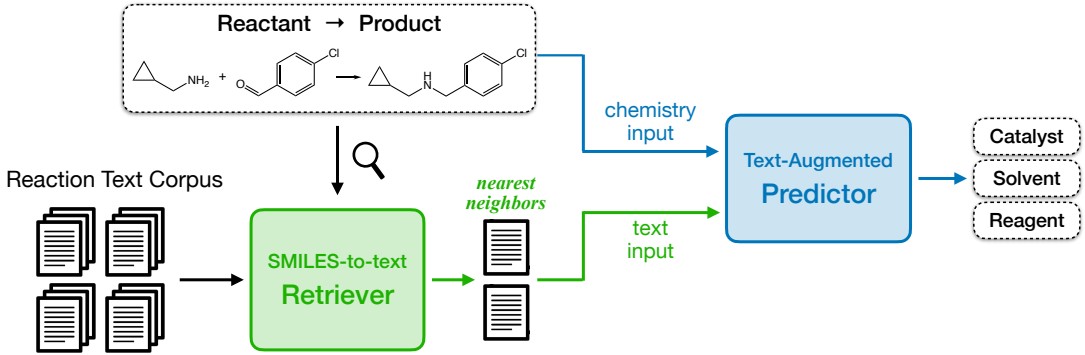

Figure 3: Overview of TextReact as applied to reaction condition recommendation. The retriever searches for texts relevant to the chemistry input, which are then used to augment the input of the predictor.

forcement learning and robotics, natural language instructions are transformed into executable actions of the agent (Branavan et al., 2009; Narasimhan et al., 2015; Ichter et al., 2022) and enable adaptability to new tasks (Hill et al., 2020).

In the chemistry domain, our work complements ongoing efforts in the automatic extraction of structured reaction data from the chemistry literature (Lowe, 2012; Swain and Cole, 2016; Guo et al., 2021; Qian et al., 2023a,b; Wilary and Cole, 2021; He et al., 2021; Nguyen et al., 2020; Vaucher et al., 2020; Zhai et al., 2019). While these works focus on creating structured databases used for training molecular models, we focus on directly leveraging unstructured natural language descriptions to further augment their predictions.

## 3 Method

### 3.1 Problem Setup

For concreteness, consider the task of reaction condition recommendation, where the input $X$ is a chemical reaction and the output $Y$ is a list of reaction conditions, including the catalyst, solvent, and reagent. We aim to train a machine learning model $\mathcal{F}$ to generate the prediction, i.e., $Y = \mathcal{F}(X)$. The model $\mathcal{F}$ is typically trained on a set of labeled training data $\mathcal{D}_{\text{train}} = \{(x_i, y_i), i = 1, \ldots, N\}$.

In this paper, we incorporate two additional resources for the training of model $\mathcal{F}$:

(1) Each example in the training set is paired with a text paragraph, i.e. $(x_i, y_i, t_i)$, where $t_i$ describes the corresponding chemical reaction. Many reaction databases are curated from the chemistry literature and provide text references for their reaction data.[2]

[2]Examples of such text references are available in the appendix.

(2) An unlabeled text corpus of chemical reactions $\mathcal{T} = \{t_j, j = 1, \ldots, M\}$, where each $t_j$ is a paragraph describing a chemical reaction. They can be easily obtained from the text of journal articles and patents using a model or heuristics to determine that they contain a reaction description.

More generally, other predictive chemistry tasks can use the same problem setup while changing the definitions of $X$ and $Y$ accordingly. For instance, in one-step retrosynthesis, $X$ is a product molecule and $Y$ is a list of potential reactants.

### 3.2 TextReact Framework

We propose a novel framework TextReact to augment chemistry prediction with text retrieval. As illustrated in Figure 3, the TextReact framework comprises two major components: a SMILES-to-text retriever (Section 3.2.1) and a text-augmented predictor (Section 3.2.2). The retriever searches the unlabeled corpus for texts relevant to the particular chemistry input. The predictor leverages both the chemistry input and the retrieved texts to generate the prediction.

### 3.2.1 SMILES-To-Text Retriever

The goal of the retriever is to locate relevant texts from an unlabeled corpus based on a given chemistry input. To accomplish this task, we devise a SMILES-to-text retriever, leveraging the widely used dual encoder architecture employed in document retrieval (Lee et al., 2019; Karpukhin et al., 2020) and image-text retrieval (Radford et al., 2021). The model consists of two parts: a *chemistry encoder* for processing the input molecule or reaction represented as a SMILES string, and a *text encoder* for encoding the text descriptions. For the

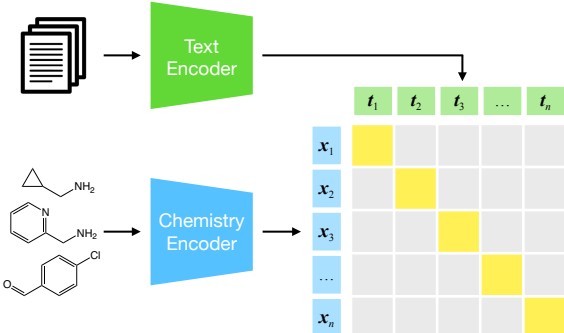

Figure 4: SMILES-to-text retriever trained with contrastive learning.

chemistry encoder, we use a Transformer to encode the SMILES string into a latent vector. As for the text encoder, we employ a Transformer pre-trained on scientific text (Beltagy et al., 2019) to encode each paragraph into a latent vector. To enable efficient retrieval using nearest neighbor search, we train the model to align the two latent spaces.

The retriever is trained with contrastive learning, as illustrated in Figure 4. Given a batch of SMILES strings and their corresponding paragraphs $\{(x_i, t_i), i = 1, \ldots, n\}$, we first compute their encodings,

$$\boldsymbol{x}_i = \text{ChemEnc}(x_i), \quad \boldsymbol{t}_i = \text{TextEnc}(t_i). \quad (1)$$

The similarity between a SMILES and a text paragraph is defined by the dot product of their encodings,

$$S_{i,j} = \boldsymbol{x}_i^\top \boldsymbol{t}_j. \quad (2)$$

For each SMILES string $x_i$, its paired paragraph $t_i$ is a positive example, and the other paragraphs within the same batch are considered negative examples. We further randomly sample $n$ paragraphs from the unlabeled corpus as additional negative examples for each batch, denoted as $t_{n+1}, \ldots, t_{2n}$. The training objective is to maximize the log-likelihood of matching the SMILES strings with positive text paragraphs, i.e.,

$$L_{\text{ret}} = -\sum_{i=1\ldots n} \log \frac{\exp(S_{i,i})}{\sum_{j=1\ldots 2n} \exp(S_{i,j})}. \quad (3)$$

After training, we pre-compute all text encodings and compile them into an index to support efficient retrieval. When retrieving relevant texts for a given SMILES string, we compute its encoding with the chemistry encoder and then run a maximum inner product search (Mussmann and Ermon, 2016; Johnson et al., 2019) to find its nearest neighbors in the index.

In this paper, we train the retriever separately from the predictor with a standalone objective. While joint optimization of the retriever and the predictor is possible (Guu et al., 2020; Lewis et al., 2020b), it requires significantly more computation as it involves iterative index rebuilding and retrieval during training. We choose a standalone retriever that has exhibited strong performance in identifying relevant texts in our experiments (see Table 2).

### 3.2.2 Text-Augmented Predictor

The goal of this component is to merge chemistry and natural language to produce accurate predictions. Our text-augmented predictor is designed as an encoder-decoder model, with the encoder handling both the chemistry input and retrieved texts, and the decoder generating the corresponding chemistry output.

The encoder's input consists of the concatenation of the chemistry input (i.e., the SMILES string) and the retrieved texts. To structure the input, we prepend a [CLS] token at the beginning and use [SEP] tokens to separate the SMILES and the texts. This yields the following input format:

[CLS] SMILES [SEP] (0) TEXT$_0$ (1) TEXT$_1 \cdots$ [SEP]

TEXT$_0$, TEXT$_1$, $\cdots$ are the nearest neighbors retrieved by the SMILES-to-text retriever, which serve as additional input to augment chemistry prediction. The number of the appended nearest neighbors $k$ is a hyperparameter.

The decoder architecture is tailored to the specific predictive chemistry task. For reaction condition recommendation, we adopt the approach established in prior research (Gao et al., 2018), which generates the reaction conditions in a specific order: catalyst, solvent 1, solvent 2, reagent 1, and reagent 2. For one-step retrosynthesis, the decoder either directly generates the SMILES strings of the reactants (i.e., a *template-free* approach), or predicts the reaction template first and uses cheminformatics software to derive the reactants (i.e., a *template-based* approach).[3]

The predictor is trained via supervised learning to maximize $p(y_i \mid x_i, T_i)$, where $T_i$ is the set of retrieved texts. To encourage the predictor to leverage the information of both chemistry and text, we add an auxiliary masked language model (MLM)

---

[3]Depending on whether to use an explicit set of reaction templates, existing one-step retrosynthesis models can be categorized into *template-based* and *template-free* approaches. See Section 5.2 and Appendix A.2 for further discussion.

loss. Specifically, we randomly mask out portions of the input, either the SMILES string or the text, and add a prediction head on top of the last layer of the encoder to predict the masked tokens. Inspired by previous research on language model pre-training (Devlin et al., 2019; Lewis et al., 2020a; Joshi et al., 2020; Du et al., 2022), the masking is performed by repeatedly sampling spans of length drawn from a Poisson distribution with $\lambda = 3$, until the total number of masked tokens reaches 15% of the original input. With the MLM loss, the model learns to infer the missing part of a SMILES string from its nearest neighbor texts, and vice versa, thus encouraging it to learn the correspondence between chemistry and text. The masking also encourages the predictor not to rely on a single source of input, similar to dropout, thus improving its robustness. Finally, the training objective of the predictor is

$$L_{\text{pred}} = -\sum_i \log p(y_i \mid x_i, T_i) + \lambda_1 L_{\text{MLM}} \quad (4)$$

where $\lambda_1$ is a hyperparameter controlling the weight of the MLM loss $L_{\text{MLM}}$.

To improve the generalizability of the predictor, we employ a dynamic sampling training strategy that addresses the following important distinction between the training and testing of the predictor. During training, we have access to the *gold text* paired with each chemistry input, providing a description of the corresponding reaction. However, in testing, it is not guaranteed to retrieve the gold text. Moreover, for novel chemistry inputs lacking a corresponding text description in the existing corpus, the predictor can only use information from the nearest neighbors, which may describe similar but not identical reactions. To address this gap, we employ a random sampling policy during training to simulate novel chemistry inputs that are not present in the corpus. With probability $\alpha$, the chemistry input is augmented with $k$ random neighbors from its top-$K$ nearest neighbors, where $K > k$ and the top-$K$ neighbors are expected to cover reactions similar to the chemistry input. In the remaining cases, the model is given the gold text along with the top-$(k-1)$ neighbors (excluding the gold text) returned by the retriever. During inference, the examples are always augmented with the top-$k$ nearest neighbors.

## 4 Experimental Setup

**Text Corpus** We construct a text corpus from USPTO patent data processed by Lowe (2012).

Each paragraph in the corpus provides a description of the synthesis procedure of a chemical reaction. The corpus consists of 2.9 million paragraphs, with an average length of 190 tokens.[4] While the original dataset includes structured reaction data extracted from each paragraph, including reactants, products, and reaction conditions, our model does not rely on this extracted data but instead learns directly from the unlabeled text.

**Implementation** The SMILES-to-text retriever is implemented with Tevatron (Gao et al., 2022).[5] The chemistry encoder is initialized with Chem-BERTa (Chithrananda et al., 2020) (pre-trained on the SMILES strings from a molecule database) and the text encoder is initialized with SciBERT (Beltagy et al., 2019) (pre-trained on scientific text from the literature). We finetune a separate retriever on the training set of each chemistry task. The input to the chemistry encoder is slightly different in each task. For reaction condition recommendation, the input is the reactants and product of a reaction, while for one-step retrosynthesis, the input is only the product. In both cases, the chemistry input is represented as a SMILES string.

The text-augmented predictor employs a pre-trained SciBERT (Beltagy et al., 2019) as the encoder. We concatenate the input SMILES string with three neighboring text paragraphs ($k = 3$) by default, using [CLS] and [SEP] tokens as described in Section 3.2.2. During training, we set the random sampling ratio to $\alpha = 0.8$ for reaction condition recommendation and $\alpha = 0.2$ for one-step retrosynthesis, and the cut-off $K$ is set to 10. We analyze the effect of the hyperparameters in Section 5.3. During inference, we use beam search to derive the top predictions. More implementation details can be found in Appendix C.

**Evaluation** For each chemistry task, we evaluate TextReact under two setups. The first setup is the *random split*, where the dataset is randomly divided into training/validation/testing. This is a commonly used setup in previous chemistry research (Gao et al., 2018; Coley et al., 2017). The second and more challenging setup is the *time split*, where the dataset is split based on the patent year. We train the model with historical data from older patents

---

[4]We use the tokenizer of SciBERT: https://huggingfac e.co/allenai/scibert_scivocab_uncased.

[5]As the original Tevatron toolkit does not support different encoders and tokenizers for queries and passages, we make modifications to accommodate these requirements.

| | RCR (RS) | RCR (TS) | RetroSyn (RS) | RetroSyn (TS) |
|---|---|---|---|---|
| Train | 546,728 | 565,575 | 40,008 | 38,631 |
| Valid | 68,341 | 63,015 | 5,001 | 5,624 |
| Test | 68,341 | 54,820 | 5,007 | 5,761 |

Table 1: Statistics of datasets for reaction condition recommendation (RCR) and one-step retrosynthesis (RetroSyn). RS: random split; TS: time split.

| | R@1 | R@3 | R@10 |
|---|---|---|---|
| RCR (RS) | 70.9 | 91.6 | 97.0 |
| RCR (TS) | 75.1 | 91.8 | 96.2 |
| RetroSyn (RS) | 60.0 | 82.5 | 92.5 |
| RetroSyn (TS) | 58.0 | 81.4 | 91.5 |

Table 2: Performance of our SMILES-to-text retriever trained on each dataset. We report the Recall@{1,3,10} when retrieving from the full corpus. Scores are in %.

and test its performance on the data from newer patents. Due to the substantial differences between reactions in new patents and previous ones, it becomes more challenging for a model to generalize effectively under a time split. During testing, we compare retrieving from the full corpus (including newer patents) and retrieving only from the years used for training. Table 1 shows the statistics of the datasets for the two tasks, which are elaborated in the next section.

## 5 Experiments

### 5.1 Reaction Condition Recommendation

**Data** We follow the setup of previous research (Gao et al., 2018) to construct reaction condition datasets from the USPTO data (Lowe, 2012). The reactions with at most one catalyst, two solvents, and two reagents are kept, and reactions with conditions that occurred fewer than 100 times are excluded.[6] Each reaction is associated with a text paragraph (gold) from the patent. However, we only utilize the gold text during the training of the retriever and the predictor, and do not use it for validation or testing.

We create two splits of the dataset: RCR (RS) for *random split*, and RCR (TS) for *time split*. More

details can be found in Appendix B.

**Baselines** We implement four baselines, none of which use additional text input:

- Reaction fingerprint (rxnfp) LSTM, a reproduction of the method proposed by Gao et al. (2018). The reaction fingerprint is calculated as the difference between the product and reactant fingerprints, which is further encoded by a two-layer neural network, and an LSTM decodes the reaction conditions.

- Reaction fingerprint (rxnfp) retrieval, which uses the conditions of the most similar reactions in the training set as the prediction. Similar reactions are determined based on the $L_2$ distance of reaction fingerprints. This baseline examines the performance of a pure retrieval method on this task.

- Transformer, the most important baseline we are comparing with, which uses the same architecture as our predictor. This baseline represents the state-of-the-art model that only takes chemistry input.

- ChemBERTa (Chithrananda et al., 2020). This baseline is the same as the Transformer baseline except that the encoder is pretrained on external SMILES data. The purpose of this baseline is to demonstrate the impact of such pretraining.

**Results** Table 3 shows our experimental results for reaction condition recommendation. TextReact substantially outperforms standard chemistry models (Transformer and ChemBERTa), which are trained on reaction data without text. This significant improvement can be attributed to the strong performance of our retriever (shown in Table 2). When the retriever successfully identifies the gold text, TextReact effectively utilizes this information to make predictions. Otherwise, TextReact can also benefit from retrieving texts of similar reactions, as we will demonstrate in Section 5.3.[7]

TextReact generalizes to a more challenging time split. As reactions from the same patent are often similar, it may be easy for the model to infer the reaction conditions when similar reactions

---

[6]Gao et al. constructed their datasets from Reaxys, which are not publicly available. We use the preprocessing script of Parrot (https://github.com/wangxr0526/Parrot) to process the public USPTO data.

[7]Another noteworthy observation is that the rxnfp retrieval baseline performs comparably to Transformer, suggesting that similar reactions often share similar reaction conditions, validating the efficacy of retrieval-based methods.

|  | RCR (RS) | | | | RCR (TS) | | | |
|---|---|---|---|---|---|---|---|---|
|  | Top-1 | Top-3 | Top-10 | Top-15 | Top-1 | Top-3 | Top-10 | Top-15 |
| rxnfp LSTM (Gao et al., 2018) | 20.5 | 30.7 | 41.7 | 45.3 | 15.2 | 26.2 | 40.7 | 45.4 |
| rxnfp retrieval | 27.2 | 37.5 | 47.9 | 51.1 | 7.8 | 15.2 | 27.3 | 31.5 |
| Transformer | 30.0 | 43.8 | 56.7 | 60.5 | 18.7 | 31.8 | 47.6 | 52.7 |
| ChemBERTa | 30.3 | 44.7 | 58.0 | 62.0 | 18.7 | 31.9 | 47.6 | 52.8 |
| TextReact | **88.4** | **93.9** | **96.0** | **96.5** | **83.9** | **90.9** | **93.9** | **94.6** |

Table 3: Evaluation results for reaction condition recommendation (RCR). RS: random split; TS: time split. Scores are accuracy in %.

|  | RetroSyn (RS) | | | | RetroSyn (TS) | | | |
|---|---|---|---|---|---|---|---|---|
|  | Top-1 | Top-3 | Top-5 | Top-10 | Top-1 | Top-3 | Top-5 | Top-10 |
| *Template-free models* | | | | | | | | |
| G2G (Shi et al., 2020) | 48.9 | 67.6 | 72.5 | 75.5 | — | — | — | — |
| Transformer (Lin et al., 2020) | 43.1 | 64.6 | 71.8 | 78.7 | — | — | — | — |
| Dual-TF (Sun et al., 2020) | 53.6 | 70.7 | 74.6 | 77.0 | — | — | — | — |
| Transformer$_{tf}$ | 45.9 | 63.2 | 68.5 | 75.2 | 31.4 | 46.6 | 51.6 | 57.2 |
| TextReact$_{tf}$ | **59.5** | **72.6** | **75.9** | **80.0** | **51.0** | **64.1** | **68.1** | **72.9** |
| *Template-based models* | | | | | | | | |
| LocalRetro (Chen and Jung, 2021) | 53.4 | 77.5 | 85.9 | 92.4 | — | — | — | — |
| O-GNN (Zhu et al., 2023) | 54.1 | 77.7 | 86.0 | **92.5** | — | — | — | — |
| Transformer$_{tb}$ | 52.5 | 72.8 | 79.7 | 86.2 | 43.6 | 65.6 | 73.2 | 82.1 |
| TextReact$_{tb}$ | **68.2** | **83.7** | **88.1** | **92.5** | **68.7** | **84.5** | **88.8** | **92.8** |

Table 4: Evaluation results for one-step retrosynthesis. RS: random split; TS: time split. Scores are accuracy in %.

are present in the training set. Unsurprisingly, all baselines perform worse under the time split, highlighting the inherent difficulty of achieving such generalization. In TextReact, despite being trained only on historical data, the retriever retains high accuracy in retrieving the corresponding paragraph for reactions in the testing set (see Table 2). Thus, TextReact retains a strong overall performance by leveraging the retrieved text.

## 5.2 One-Step Retrosynthesis

**Data** We use the popular USPTO-50K dataset (Coley et al., 2017) for our one-step retrosynthesis experiment. We also create two data splits: RetroSyn (RS), the original *random split* of the dataset, and RetroSyn (TS), the *time split*.

Since the dataset was constructed from the same USPTO data as our text corpus, we match the examples in the dataset with text paragraphs in the corpus. However, due to differences in preprocessing, not all examples could be matched.[8] We use only

the matched examples to train the retriever, whereas the predictor is trained with the full dataset.

**Template-free & Template-based Models** We implement TextReact in two settings: (1) TextReact$_{tf}$ is a template-free model that uses a Transformer decoder to generate the SMILES strings of reactants. (2) TextReact$_{tb}$ is a template-based model that follows the formulation of LocalRetro (Chen and Jung, 2021). Specifically, we adopt the set of reaction templates extracted from the training data, and predict which template is applicable to a product molecule and which atom or bond is the reaction center. For each atom, we represent it using the corresponding hidden state from the last layer of the Transformer encoder and predict a probability distribution over the reaction templates using a linear head. For each bond, we concatenate the representations of its two atoms and employ another linear head to predict the template.

---

[8] Out of the 40,008 examples in the training set, 31,391 are matched.

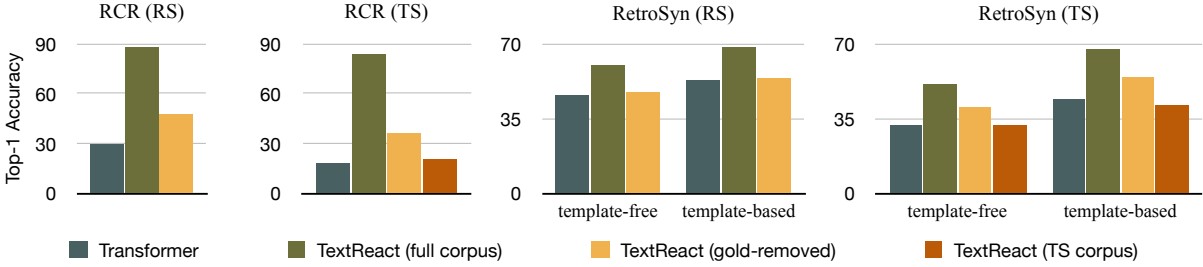

Figure 5: Comparison of TextReact's performance under three settings: (1) Retrieving from the full corpus, (2) Retrieving from the corpus with gold texts of testing examples removed, and (3) Retrieving only from the historical corpus under a time split.

**Baselines** We implement baselines Transformer$_{tf}$ and Transformer$_{tb}$ for template-free and template-based settings, respectively. They closely follow the output format of TextReact but do not incorporate retrieved text as additional input. We also report the published results of previous research (Shi et al., 2020; Lin et al., 2020; Sun et al., 2020; Chen and Jung, 2021; Zhu et al., 2023). All baselines are trained on the same data as TextReact, but without the text input.

**Results** Table 4 shows the results of one-step retrosynthesis. Similar to the RCR task, TextReact has demonstrated strong performance by leveraging text retrieval. On the RetroSyn (RS) dataset, our baseline Transformer models perform comparably to previous models under both template-free and template-based settings. Upon integrating text augmentation, TextReact$_{tf}$ and TextReact$_{tb}$ advance the top-1 accuracy by 13.6% and 15.7%, respectively, affirming the advantage of retrieval augmentation for this task. Even under a time-split scenario, TextReact maintains a high accuracy, underscoring its proficiency in both retrieval and final prediction stages.

### 5.3 Analysis

First, we demonstrate in Figure 5 that TextReact exhibits generalization capabilities to novel reactions not present within the text corpus. While we have illustrated the accurate retrieval of text descriptions from the corpus and their effective utilization for predictions, it is important to note that a gold text for the target reaction may not always be available within the corpus. To further assess the model's performance in a more challenging scenario, we remove the gold texts of all testing examples from the corpus (referred to as "gold-removed" in Figure 5). In both RCR and RetroSyn, TextReact continues to outperform the

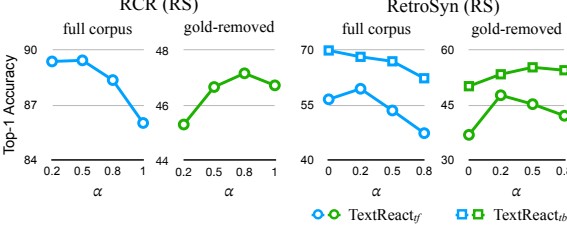

Figure 6: Performance of TextReact with respect to the random sampling ratio $\alpha$ during training.

Transformer baseline significantly. The improvement is consistent under both the random split and time split, albeit smaller as compared with retrieving from the full corpus. However, we note that under the time split, if the model is only allowed to retrieve from the historical corpus (the patents for the training data, referred to as "TS corpus"), TextReact can hardly outperform the baseline. The reason behind this could be that the reactions in new patents are sufficiently different from those in historical patents, placing a predictive barrier for such out-of-distribution generalization.

TextReact's robust generalization performance is enabled by the sampling strategy we adopted during training. Figure 6 illustrates the impact of the random sampling ratio $\alpha$. During training, the predictor is given the gold text paragraph with probability $1 - \alpha$, and randomly sampled paragraphs with a probability $\alpha$. The results reveal that the choice of $\alpha$ plays an important role in the model, both when retrieving from the full corpus and the gold-removed corpus. In the RCR (RS) dataset, the full corpus evaluation prefers a smaller $\alpha$, while the gold-removed evaluation prefers a larger $\alpha$. Since $\alpha$ controls the probability of using gold input in training, a larger $\alpha$ aligns better with the gold-removed setting. We set $\alpha = 0.8$ for the RCR experiments for a good balance between the two settings. In RetroSyn (RS), however, the model

| | RCR (RS) | | RetroSyn (RS) | |
|---|---|---|---|---|
| | full | g.r. | full | g.r. |
| TextReact | 88.4 | 47.2 | 59.5 | 47.7 |
| · SMILES-only | 30.0 | 30.0 | 45.9 | 45.9 |
| · text-only | 81.7 | 38.3 | 23.0 | 9.0 |
| · no MLM | 87.7 | 47.0 | 51.0 | 43.5 |
| · no pretrain | 83.5 | 43.8 | 48.1 | 40.3 |
| · sep. neighbors | 80.7 | 45.2 | 49.4 | 45.3 |

Table 5: Ablation study of TextReact. We evaluate TextReact$_{tf}$ for the RetroSyn task. (full: full corpus; g.r.: gold-removed)

favors a much smaller $\alpha$. We hypothesize that this difference is due to the varying helpfulness of the retrieved texts in different tasks. The texts appear to be more helpful for condition recommendation than retrosynthesis, perhaps due to the existence of common reaction conditions that can be reused across many different reactions of the same type.

Table 5 presents our ablation study. First, TextReact performs significantly better than the model that uses only the input SMILES or retrieved text, illustrating that TextReact effectively integrates both chemistry and text inputs to generate its predictions. Second, we observe a significant increase in accuracy for TextReact compared to the model trained without the MLM loss. This demonstrates the effectiveness of the auxiliary MLM objective in enhancing the model's learning of the correspondence between chemistry and text inputs. In addition, TextReact benefits from the SciBERT checkpoint, which has been pretrained on scientific text. While TextReact concatenates the top-$k$ neighbors together to make predictions, we compare with a variant that separates the neighbors and ensembles their predictions, similar to RAG (Lewis et al., 2020b). This variant yields worse performance (the last row of Table 5), suggesting the benefits of jointly encoding the neighbors.

Additional analyses in Appendix D reveal several key findings: (1) TextReact achieves superior performance on the RCR task using only 10% of the training data compared to the Transformer baseline; (2) TextReact performs better when the retrieved texts describe reactions that bear closer similarity to the input reaction; and (3) TextReact benefits from jointly modeling the input reaction and retrieved neighboring texts.

## 6 Conclusion

This paper presents TextReact, a novel method that augments predictive chemistry with text retrieval. We employ information retrieval techniques to identify relevant text descriptions for a given chemistry input from an unlabeled corpus, and supply the retrieved text as additional evidence for chemistry prediction. In two chemistry tasks, TextReact demonstrates strong performance when retrieving from the full corpus, and maintains a significant improvement when retrieving from a harder corpus that excludes the gold texts.

Our results highlight the promising potential of incorporating text retrieval methods in the field of chemistry. As chemically similar reactions have similar conditions and outcomes, effectively retrieving and grounding textual knowledge from patents and publications into the chemistry space can significantly enhance the predictive power of computational models.

## 7 Limitations

We acknowledge two limitations of this paper. First, our experiments focused on two representative chemistry tasks, but we believe that the proposed method can be applied to other tasks and domains that would benefit from the knowledge in the literature. Second, we employed a simplified implementation of the retriever and predictor models to demonstrate the effectiveness of retrieval augmentation in chemistry. There is significant room for further improvements, such as using more advanced pretrained models and exploring joint training of the models.

## 8 Acknowledgements

The authors thank Wenhao Gao, Xiaoqi Sun, Luyu Gao, and Vincent Fan for helpful discussion and feedback. This work was supported by the NSF Expeditions grant (award 1918839: Collaborative Research: Understanding the World Through Code), the Machine Learning for Pharmaceutical Discovery and Synthesis (MLPDS) consortium, the Abdul Latif Jameel Clinic for Machine Learning in Health, the DTRA Discovery of Medical Countermeasures Against New and Emerging (DOMANE) threats program, the DARPA Accelerated Molecular Discovery program, the NSF AI Institute CCF-2112665, the NSF Award 2134795, the GIST-MIT Research Collaboration grant, MIT-DSTA Singapore collaboration, and MIT-IBM Watson AI Lab.

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

## A Background

### A.1 Reaction Condition Recommendation

Reaction condition recommendation is the task of suggesting reaction conditions, such as catalysts, solvents, and reagents, for a chemical reaction. Gao et al. (2018) proposed a machine learning model for this task, where the reaction is represented as the difference between the product and the reactant fingerprint vectors, indicating the change of substructures during the reaction, and the reaction conditions are sequentially predicted by the model. Later research studied different methods to represent the reaction, such as MACCS key fingerprints (Walker et al., 2019) and graph neural networks (Ryou et al., 2020), and alternative machine learning formulations, such as multilabel classification (Maser et al., 2021) and variational inference (Kwon et al., 2022). In this work, we follow Gao et al.'s formulation but use the more advanced Transformer architecture.

### A.2 One-Step Retrosynthesis

Another important task in predictive chemistry is one-step retrosynthesis, which aims to propose reaction precursors (reactants) for target molecules (products). Prior methods can be broadly classified into two categories: *template-based* and *template-free*. Template-based approaches use classification models to predict the reaction template that can be applied to the target (Segler and Waller, 2017; Chen and Jung, 2021), and subsequently employ cheminformatics software to derive the precursors based on the template. Template-free approaches have a

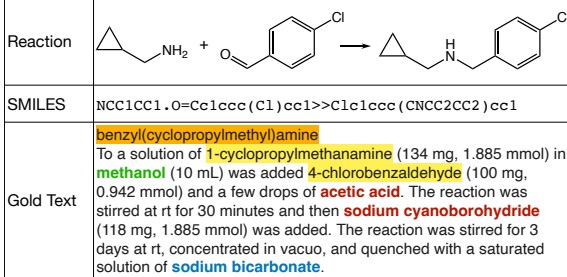

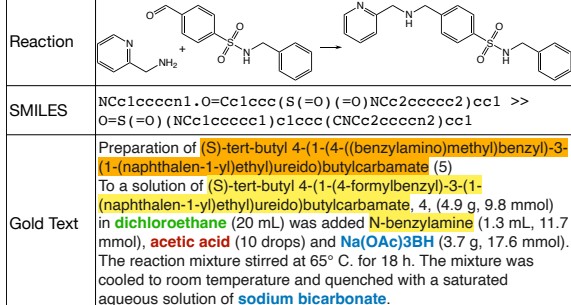

Figure 7: Example reactions and their corresponding gold texts in the USPTO data. Reactants and products are marked in yellow and orange, while the catalyst, solvent, and reagent are displayed in red, green, and blue, respectively.

more ambitious goal: predicting the precursors directly without relying on a fixed set of reaction templates. Such approaches are usually implemented as graph-to-graph or sequence-to-sequence generation models (Liu et al., 2017; Dai et al., 2019; Shi et al., 2020; Lin et al., 2020; Somnath et al., 2021), and achieve comparable performance to template-based approaches on common benchmarks. In this work, we apply TextReact to both template-based and template-free approaches.

## B Datasets

Figure 7 shows two example reactions and their corresponding gold texts in the USPTO corpus.

**Reaction Condition Recommendation** The dataset is constructed from the public USPTO data (Lowe, 2012).[9] We create two splits:

- RCR (RS), where the reactions are *randomly split* into training/validation/testing with a ratio of 80%/10%/10%.

- RCR (TS), where we perform a *time split* of the dataset. The reactions collected from patents before 2015 are categorized as the

---

[9]https://doi.org/10.6084/m9.figshare.5104873.v1

training set, reactions from 2015 as validation, and reactions from 2016 as testing.

**One-Step Retrosynthesis** We adopt the USPTO 50K dataset (Coley et al., 2017)[10], which is commonly used in previous works, and create two splits:

- RetroSyn (RS), i.e. *random split*, which is the original split of the dataset.

- RetroSyn (TS), i.e. *time split*, where we merge the original training, validation, and testing sets and re-split based on the patent year. Data before 2012 are used for training, data from 2012 and 2013 are used for validation, and data from 2014 and 2015 are used for testing.

Our datasets will be publicly available to foster future research in this direction.

## C Implementation Details

Our experiments are implemented with the Hugging Face Transformers (Wolf et al., 2020), PyTorch Lightning[11], Tevatron (Gao et al., 2022), and Faiss (Johnson et al., 2019) libraries.

**Reaction Condition Recommendation** The SMILES-to-text retriever consists of a chemistry encoder and a text encoder. We train the retriever by contrastive learning, where each batch contains 512 queries (chemistry inputs), and each query is associated with one positive paragraph and one random sampled negative paragraph from the corpus. Both query and paragraph have a maximum length of 256. We train the retriever for 50 epochs using a learning rate of $1e-4$ (with 10% warmup and linear decay).

The predictor is trained for 20 epochs using a batch size of 128 and a learning rate of $1e-4$ (with 2% warmup and cosine decay). The weight of the MLM loss is $\lambda_1 = 0.1$. We apply data augmentation by generating the SMILES strings with a random order during training. By default, we append the chemistry input with $k = 3$ neighboring text paragraphs, and a maximum length of the encoder input is 512. The input SMILES string and text are tokenized with the same SciBERT tokenizer. For the experiments with $k = 1, 2, 4, 5$, we set the maximum length to 256, 384, 768, 1024, respectively.

[10] https://github.com/coleygroup/openretro/tree/main/data/USPTO_50k

[11] https://www.pytorchlightning.ai/index.html

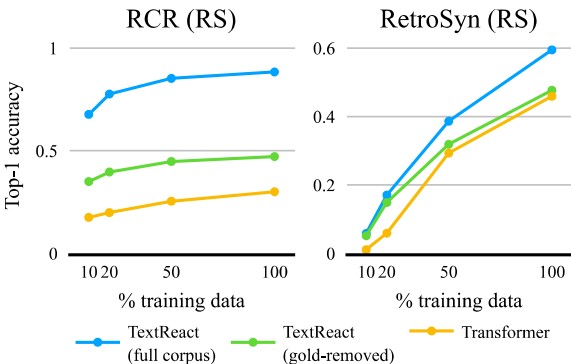

Figure 8: Top-1 accuracies of TextReact and the Transformer baseline vs. the amount of training data used, as a percentage of the total available training data. For RCR, TextReact exhibits strong performance even in low-resource scenarios. For RetroSyn, we evaluate TextReact$_{tf}$ in this figure.

**One-Step Retrosynthesis** The implementation and hyperparameters for the one-step retrosynthesis experiments largely resemble those of the reaction condition recommendation experiments, with a few differences. The retriever is trained for 400 epochs due to the smaller size of the RetroSyn (i.e., USPTO 50K) dataset. Each query (the SMILES string of the product molecule) has a maximum length of 128.

For the predictor, the template-free TextReact$_{tf}$ and template-based TextReact$_{tb}$ architectures are slightly different. While both employ a SciBERT encoder, TextReact$_{tf}$ uses a 6-layer Transformer decoder to generate the output SMILES string, and TextReact$_{tb}$ predicts the reaction template using linear heads on atom and bond representations. In both scenarios, the predictor is trained for 200 epochs. The random sampling ratio in training is also set differently ($\alpha = 0.2$) as mentioned in Section 4.

## D Further Analysis

To investigate the effect that the amount of training data has on model performance, we trained TextReact in both the RCR and RetroSyn on 10%, 20%, 50%, and 100% of the available training data under the random split. The same was done with the Transformer baseline model for comparison. The models are evaluated on the full test dataset and the top-1 accuracies are plotted in Figure 8. As expected, both TextReact and the Transformer baseline see a noticeable improvement in performance as the amount of training data is increased. In the RCR domain, the full-corpus accuracy and

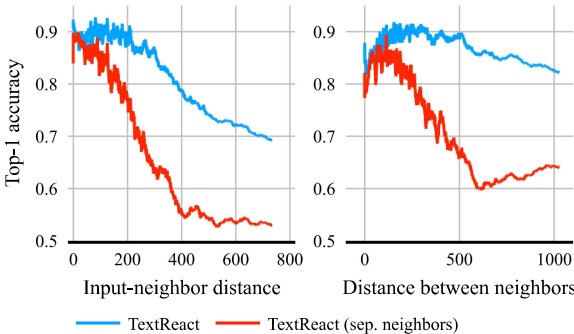

Figure 9: Top-1 accuracies of TextReact and TextReact (sep. neighbors) on RCR (RS) when retrieving from the full corpus, plotted against the average input-neighbor distance and the average distance between neighbors.

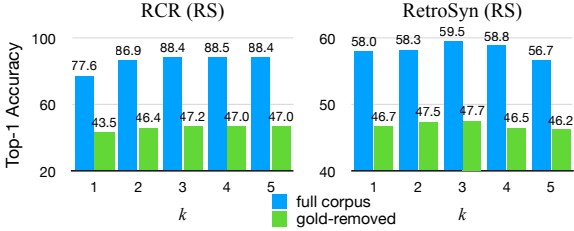

Figure 10: Performance of TextReact with respect to the number of neighbors $k$ retrieved for each input. We evaluate TextReact$_{tf}$ for RetroSyn.

gold-removed accuracy of TextReact when trained on 10% of the available training data are 67.7% and 35.0%, respectively, both of which are higher than the accuracy of the baseline as trained on the *full training set* (30.0%). This shows that our model is capable of achieving better performance with only 10% as much training data for reaction condition recommendation.

To study the effect that the retrieved texts have on model performance, we plotted the top-1 accuracy of TextReact and TextReact (sep. neighbors) with respect to the *average input-neighbor distance*, defined as the average $L_2$ distance between the fingerprint of the input reaction and the fingerprints of the reactions corresponding to the retrieved texts (left panel of Figure 9). This distance measures the difference between the input reaction and the reactions described by the retrieved texts. The plot shows that model predictions are more accurate when the retrieved texts correspond to reactions more similar to the input reaction, thus confirming that TextReact effectively incorporates information from the retrieved texts when making predictions.

We also plotted the accuracy with respect to the *average distance between neighbors*, defined as the average $L_2$ distance between the fingerprints

of the reactions corresponding to each pair of retrieved texts (right panel of Figure 9). This measures how different the retrieved neighbors are from each other. The graph shows that TextReact's performance does not depend much on how similar the retrieved neighbors are to each other, whereas the performance of TextReact (sep. neighbors) drops when the retrieved neighbors are more different from each other. When separately encoding the neighbors, the model does not know how much it should trust each neighbor. On the other hand, TextReact does not suffer from the same performance drop, likely due to the attentions between the retrieved texts, which allow the model to better integrate the information from the neighbors. This demonstrates the benefits of concatenating the input reaction and all the retrieved texts together and feeding them into a single Transformer encoder.

Figure 10 analyzes the number of text paragraphs $k$ that we retrieve for each example. In the RCR (RS) dataset, there is a notable improvement from $k = 1$ to $k = 3$, suggesting the benefits of retrieving more neighbors as additional context. In the RetroSyn (RS) dataset, the trend is similar but the improvement is smaller. However, the performance decreases when $k > 3$ for both datasets. Therefore, we set $k = 3$ for our main experiments.