# OpenReview forum: "Predictive Chemistry Augmented with Text Retrieval"
_EMNLP/2023/Conference — EMNLP 2023 Main_

### Official Review · Reviewer_3bHy · 2023-07-22

**Soundness:** 3

**Excitement:**

3: Ambivalent: It has merits (e.g., it reports state-of-the-art results, the idea is nice), but there are key weaknesses (e.g., it describes incremental work), and it can significantly benefit from another round of revision. However, I won't object to accepting it if my co-reviewers champion it.

**Paper Topic And Main Contributions:**

This paper proposes a new retrieval-augmented approach in the chemistry domain. Specifically, the proposed model retrieves relevant paragraphs to a given input SMILES string from a patent corpus, USPTO. A cross-encoder then encodes textual and chemical data and a decoder predicts the target SMILES string. The proposed model is evaluated on two downstream tasks, reaction condition recommendation and one-step retrosynthesis, and outperforms simple single-modal baselines that use only SMILES strings as features.

**Reasons To Accept:**

1. The paper is easy to follow and is nicely read in most parts.

2. The proposed retriever may benefit other practitioners and researchers in the field of chemistry if it is open-sourced.

**Reasons To Reject:**

1. The effectiveness of the proposed model has not been sufficiently demonstrated because state-of-the-art models were not compared in the experiments. In the one-step retrosynthesis task, while several previous models exhibit superior performance than the performance of the proposed model, they were not considered in the conducted experiments (Table 4). Please review previous studies [1,2] and this link [3] to ensure that essential baselines have not been overlooked.

If the authors feel that it is unfair to compare the previous best models I have mentioned with the proposed model (i.e., TextReact), please reply during the author response period. I will change the initial score if the responses resolve my concerns.

[1] Zhu et al., O -GNN: incorporating ring priors into molecular modeling, ICLR 2023

[2] Seidl et al., Improving few-and zero-shot reaction template prediction using modern hopfield networks, Journal of chemical information and modeling, 2022

[3] https://paperswithcode.com/sota/single-step-retrosynthesis-on-uspto-50k

**Reproducibility:**

3: Could reproduce the results with some difficulty. The settings of parameters are underspecified or subjectively determined; the training/evaluation data are not widely available.

**Reviewer Confidence:**

3: Pretty sure, but there's a chance I missed something. Although I have a good feel for this area in general, I did not carefully check the paper's details, e.g., the math, experimental design, or novelty.

---

> ### Author Rebuttal · Authors · 2023-08-28
>
> We appreciate the reviewer for their valuable comments. Below, we provide answers to the reviewer's questions.
>
> **Q1. State-of-the-art models were not compared in the experiments.**
>
> First, we would like to thank the reviewer for pointing out additional models that were absent from our current results. Table 3 for reaction condition recommendation has covered state-of-the-art models.  We will extend Table 4 to include additional models, and also cite and discuss relevant papers. The following table shows the expanded results on RetrySyn (RS), i.e., the USPTO 50K dataset.
>
> | | Template-Free | Top-1 | Top-3 | Top-5 | Top-10 |
> |--|--|--|--|--|--|
> | G2G (Shi et al., 2020) | Yes | 48.9 | 67.6 | 72.5 | 75.5 |
> | Transformer (Lin et al., 2020) | Yes | 43.1 | 64.6 | 71.8 | 68.7 |
> | Transformers (ours) | Yes | 45.9 | 63.2 | 68.5 | 75.2 |
> | Dual-TF | Yes | 53.6 | 70.7 | 74.6 | 77.0 |
> | LocalRetro | No | 53.4 | 77.5 | 85.9 | 92.4 |
> | O-GNN | No | 54.1 | 77.7 | 86.0 | 92.5 |
> | TextReact | Yes | 59.5 | 72.6 | 75.9 | 80.0 |
>
> We can see that TextReact, as reported in the paper, outperforms state-of-the-art template-free models, and also outperforms state-of-the-art template-based models in terms of top-1 accuracy. Template-based models which were excluded from the current paper adopt a different pipeline. They first predict the reaction templates that can be applied to synthesize the product, and then use rule-based cheminformatics tools to compute the reactant SMILES (see Appendix A). These models can achieve better accuracy but rely on additional resources, such as reaction templates and cheminformatics tools. Besides, they are fundamentally incapable of generalizing to new reaction templates [1] [2].
>
> The main purpose of our paper is to demonstrate how text retrieval can augment different chemistry prediction tasks, rather than aiming to surpass the state-of-the-art on one particular task. Thus we did not include template-based models in our previous experiment (see lines 458-470).
> Our ongoing work is investigating retrieval augmentation within a template-based approach as well, and we will include further discussion in the paper. Nevertheless, we remain confident that our experiments have adequately demonstrated the advantages of integrating text retrieval into chemistry tasks. We hope our response has addressed the reviewer's concerns.
>
> Finally, we will release our code and data to enable reproduction of the experiments and to support future research.
>
> [1] Lin, Kangjie, et al. "Automatic retrosynthetic route planning using template-free models." Chemical science (2020).
>
> [2] Tu, Zhengkai, et al. "Predictive chemistry: machine learning for reaction deployment, reaction development, and reaction discovery." Chemical Science (2023).

---

### Official Review · Reviewer_zvNB · 2023-08-01

**Soundness:** 4

**Excitement:**

3: Ambivalent: It has merits (e.g., it reports state-of-the-art results, the idea is nice), but there are key weaknesses (e.g., it describes incremental work), and it can significantly benefit from another round of revision. However, I won't object to accepting it if my co-reviewers champion it.

**Missing References:**

Text2Mol: Cross-Modal Molecule Retrieval with Natural Language Queries

**Paper Topic And Main Contributions:**

This paper introduces TextReact, a retrieval-based solution that predicts the solvents, reagents, and catalysts of chemical reactions by retrieving relevant text portions from a Patent dataset.

TextReact formulates the task as a bi-encoder-based retrieval from text given a multi-model molecular representation. To Motivate their work the authors generate a novel chemical retrieval benchmark and showcase the utility of their model.

**Reasons To Accept:**

1. TextReact allows for accurate prediction of reaction properties with minimal overhead.
2. This work will likely inspire novel research into retrieval augmented reaction prediction.

**Reasons To Reject:**

1. The paper does not study any additional methods but bi-encoders. The contribution would significantly be improved by integrating a cross-encoder or sparse retriever like BM25.
2. It is unclear how the model performs in more difficult molecular reactions where there can be one, many or no catalysts or solvents
3. Retrieval metrics are missing such as recall@100

**Reproducibility:**

4: Could mostly reproduce the results, but there may be some variation because of sample variance or minor variations in their interpretation of the protocol or method.

**Reviewer Confidence:**

4: Quite sure. I tried to check the important points carefully. It's unlikely, though conceivable, that I missed something that should affect my ratings.

---

> ### Author Rebuttal · Authors · 2023-08-28
>
> We appreciate the reviewer for their valuable comments. Below, we provide answers to the reviewer's questions.
>
> **Q1. The paper does not study any additional methods but bi-encoders.**
>
> Our method is designed to address a multimodal retrieval task, i.e., retrieving texts related to a given chemical reaction. Bi-encoder (termed as dual encoder in our paper) is a commonly adopted model in this context. As discussed in our related work section, our approach is similar to CLIP, a bi-encoder capable of retrieval between images and texts.
>
> Regarding the two methods mentioned by the reviewer,
> - Cross-encoders are not ideally suited for retrieval tasks; instead, they are usually used for reranking a limited number of candidates. As a cross-encoder must jointly encode the query and each document, it is computationally intractable for large corpora. In contrast, bi-encoders are better suited as the encodings of documents can be pre-computed separately. Given the strong performance achieved by our bi-encoder retriever (see Table 2), we chose not to train another cross-encoder to rerank the retrieved candidates.
> - Sparse retrievers such as BM25 excel in text-to-text retrieval tasks, but cannot be trivially applied to our cross-modal SMILES-to-text retrieval task. It is because the two modalities do not have a shared vocabulary. Sparse retrievers rely on token matching, a simple and effective inductive bias for measuring similarity in pure text domains. However, when the query is a SMILES string and the document is natural language text, token matching does not work well because they do not share the same vocabulary. Thus, we did not adopt a sparse retriever.
>
> **Q2. It is unclear how the model performs in more difficult molecular reactions where there can be one, many, or no catalysts or solvents.**
>
> We would like to clarify that our reaction condition recommendation task has considered reactions with one, many or no catalysts or solvents. Specifically, we follow the exact setup of previous research (Gao et al., 2018) to construct a reaction dataset with at most one catalyst, two solvents, and two reagents (section 5.1), which covers reactions without catalyst, solvent, or reagent. The maximum number of catalysts/solvents/reagents is determined based on the majority of the USPTO reaction data (the same as Gao et al., 2018). Our predictor generates catalysts/solvents/reagents in a sequence, and is able to handle a varied number of conditions. For example, when a reaction doesn’t have a catalyst, the predictor generates a “None” for that component.
>
> **Q3. Retrieval metrics are missing such as recall@100**
>
> Our retrieval metrics are reported in Table 2, including recall@{1,3,10}. However, recall@100, as specifically mentioned by the reviewer, is not included here since we do not seek to retrieve that many texts for a given reaction. Our main experiments use 3 retrieved paragraphs as additional input. As shown in the additional analysis in Appendix D Figure 10, we have conducted experiments using at most 5 retrieved paragraphs, and retrieving more texts is not always helpful.
>
> **Q4. Missing reference**
>
> We thank the reviewer for pointing out the missing reference to Text2Mol, which retrieves molecules using natural language queries. Our retriever operates in the opposite direction, retrieving natural language descriptions using molecules, and we further use the retrieved text to enhance chemistry prediction tasks. Both papers contribute to the application of NLP techniques in chemistry. We will ensure to cite Text2Mol in our final version.

---

### Official Review · Reviewer_EhYp · 2023-08-04

**Soundness:** 4

**Excitement:**

4: Strong: This paper deepens the understanding of some phenomenon or lowers the barriers to an existing research direction.

**Paper Topic And Main Contributions:**

The paper proposes to make use of textual side information in machine learning models for predicting the outcome of chemical reactions. Textual side information is retrieved from chemical literature on the basis of the specifications of the chemical species (SMILES). This is done by learning a joint embedding space and kNN retrieval in the spirit of the popular image/text model CLIP. Then, both the original SMILES description and the retrieved natural language data are used together to make a prediction.

The main claim is that this natural-language-augmentation improves the performance on various chemical prediction tasks, such as condition recommendation and one-step retrosynthesis.

**Questions For The Authors:**

A. On a high-level, I wonder how useful the text augmentation is in practice? (With my limited background in chemistry), I would assume that these models are mainly used to predict previously unknown outcomes. Yet anything described in the literature can be hardly considered unknown. As you already explored a setting where the "gold texts" are removed, I assume the advantage then stems from finding the known effects of e.g. components in the literature?

B. Are there really no available baselines that employ text retrieval, or perform the same tasks with text only?

**Reasons To Accept:**

The experiments provide strong support for their claim, achieving more than twice as high accuracy compared to the best baseline ChemBERTa.

The paper further considers two different types of train/test splits (random and time-based). The advantage of the proposed method is confirmed in both scenarios.

Claims are further supported through careful ablations, such as testing a setting where the gold texts are removed and a setting where only the historical corpus is used in the time split setting. The method still shows a solid advantage over a Transformer baseline (which was ~on par with ChemBERTa in other experiments).

**Reasons To Reject:**

The paper could be further improved by analyzing the influence of SMILES input vs. retrieved text input. Given the comparison with the literature that only uses SMILES input, it seems that retrieved text provides an extremely rich source of information. It leads to the question how a model would perform only based on retrieved text, if feasible.

The paper would benefit from a discussion of how useful such models are in practice (see Question).


**Reproducibility:**

4: Could mostly reproduce the results, but there may be some variation because of sample variance or minor variations in their interpretation of the protocol or method.

**Reviewer Confidence:**

3: Pretty sure, but there's a chance I missed something. Although I have a good feel for this area in general, I did not carefully check the paper's details, e.g., the math, experimental design, or novelty.

**Typos Grammar Style And Presentation Improvements:**

- define the acronym SMILES
- Introduction "we simulate novel inputs by eliminating from the training data the closest textual descriptions for given reactions." could already be more clear about whether this is training strategy or an evaluation strategy.

---

> ### Author Rebuttal · Authors · 2023-08-28
>
> We appreciate the reviewer for their valuable comments. Below, we provide answers to the reviewer's questions.
>
> **Q1. The influence of SMILES input vs. retrieved text input.**
>
> Thank you to the reviewer for this valuable suggestion! In response, we have conducted an additional ablation study using a model that solely utilizes the retrieved text as input. The results are presented below, displaying the Top-1 accuracy in %. (RCR: reaction condition recommendation. RetroSyn: retrosynthesis.)
>
> | |   RCR (RS)  | RetroSyn (RS) |
> |-| :---------: | :------------: |
> | TextReact |   88.4   |   59.5   |
> | Text-only |  81.7  |  23.0  |
> | SMILES-only | 30.0 | 45.9 |
>
> We observe that the text-only model performs less effectively than TextReact on both tasks, illustrating that TextReact effectively utilizes both chemistry and text inputs to generate its predictions. We also note an intriguing distinction when comparing the text-only and SMILES-only models. The former demonstrates better performance in RCR, likely due to the explicit textual expression of reaction conditions in retrieved text. Conversely, the latter outperforms in RetroSyn, showing that text information offers less utility in this context.
>
> **Q2. How useful is text augmentation in practice? I would assume these models are mainly used to predict previously unknown outcomes.**
>
> As the reviewer pointed out, it is important to evaluate predictive chemistry models on previously unknown outcomes, i.e., those not covered in existing literature. We have two ways to validate that TextReact generalizes in such scenarios: evaluation on unknown reactions (“gold-removed” setup in Figure 5), and a time-based split (Table 3, 4 and Figure 5). These results demonstrate that our model does not just extract answers from text, but leverages similar reaction data to enhance its prediction.
>
> The generalization capability of TextReact can be attributed to two reasons: (1) chemically similar reactions have similar conditions and outcomes; (2) effective grounding enabled by jointly modeling text and chemistry formulas. In practice, TextReact can predict unknown outcomes by retrieving similar reactions as additional evidence. Moreover, as new publications emerge, TextReact can seamlessly retrieve raw texts from these new publications to improve the prediction on newly-discovered reactions (as demonstrated by our time-split experiments).
>
> We will definitely include more discussions about how TextReact could be used in practice and its benefits, as suggested by the reviewer.
>
> **Q3. Are there really no baselines that employ text retrieval, or perform the same tasks with text only?**
>
> Chemistry prediction tasks, like reaction condition recommendation and one-step retrosynthesis, have traditionally been studied through pure chemistry-based models, e.g., encoding molecules and reactions with neural networks to facilitate predictions. To the best of our knowledge, there is no previous work leveraging text retrieval for these specific tasks. Similarly, we are not aware of text-only models for these tasks as the input is a chemical reaction, not text. However, we trained another model with retrieved text only as additional ablation study, as detailed in our response to Q1.

---

### Meta-Review · Area_Chair_9XdG · 2023-10-04

**Recommendation:** 4

**Metareview:**

The authors describe a methodology, TextReact, for including retrieved relevant textual information with respect to a particular chemistry input (e.g., molecular encodings)for predicting chemical reaction tasks (e.g., reaction condition recommendation, one-step retrosynthesis) using machine learned models . Specifically, they use a SMILES-to-text retriever to retrieve relevant text, which is then also encoded (with the chemistry input) to be used as input to a machine learned model -- in this case an encoder-decoder model. Experiments are conducted  using a patent corpus as the auxiliary text source for the reaction condition recommendation and one-step retrosynthesis tasks, showing strong performance over existing baseline methods and a transformer network without text augmentation.

== Quality == The reviewers agreed that the paper was well-written overall, that the methodological approach is well-motivated in considering recent NLP advances, and that the experimental results follow a rigorous procedure and demonstrate strong performance. In combination with the primary results, the the ablation studies further confirm the results, making for a sufficiently convincing case for TextReact as a useful method. The two concerns in this regard were that reviewer EhYp had a question regarding 'retrieved text-only' and reviewer 3bHy pointed out that the experiments aren't universally SotA for both tasks depending on the configuration. However, the authors rebutted this well -- some of these results and discussion which should likely be included in the final version of the paper as they are valid questions.

== Clarity == As previously stated, the paper is well-written overall, especially considering that this isn't a chemistry knowledgeable community (in general). In particular, the figures are effective in giving an overview and providing a scaffolding for the structure of the document. The one comment made is that some qualitative analysis would be useful to given an intuitive perspective regarding characterizing the utility of the text-augmentation in specific settings.

== Originality == As far as the reviewers and I can tell, this is the first use of (text) retrieval-augmented chemistry related predictions. It is a relatively simple augmentation approach shown to work well in practice for two important tasks.

== Significance = First of all, the empirical results are quite promising and convincing -- validating the method and making chemistry contributions in some cases. As RAG-style systems have gained increasing popularity, it isn't unreasonable to believe that this may also be able to be integrated into general LLMs and this work code be a basis for such integration. Overall, this is a good case of applying NLP technology to other potentially impactful use cases. As the code is promised to be released, it is also lowering the barrier for others to continue work in this direction.

---

### Decision · Program_Chairs · 2023-10-07

**Decision:**

Accept-Main

**Comment:**

The authors describe a methodology, TextReact, for including retrieved relevant textual information with respect to a particular chemistry input (e.g., molecular encodings)for predicting chemical reaction tasks (e.g., reaction condition recommendation, one-step retrosynthesis) using machine learned models . Specifically, they use a SMILES-to-text retriever to retrieve relevant text, which is then also encoded (with the chemistry input) to be used as input to a machine learned model -- in this case an encoder-decoder model. Experiments are conducted  using a patent corpus as the auxiliary text source for the reaction condition recommendation and one-step retrosynthesis tasks, showing strong performance over existing baseline methods and a transformer network without text augmentation.

== Quality == The reviewers agreed that the paper was well-written overall, that the methodological approach is well-motivated in considering recent NLP advances, and that the experimental results follow a rigorous procedure and demonstrate strong performance. In combination with the primary results, the the ablation studies further confirm the results, making for a sufficiently convincing case for TextReact as a useful method. The two concerns in this regard were that reviewer EhYp had a question regarding 'retrieved text-only' and reviewer 3bHy pointed out that the experiments aren't universally SotA for both tasks depending on the configuration. However, the authors rebutted this well -- some of these results and discussion which should likely be included in the final version of the paper as they are valid questions.

== Clarity == As previously stated, the paper is well-written overall, especially considering that this isn't a chemistry knowledgeable community (in general). In particular, the figures are effective in giving an overview and providing a scaffolding for the structure of the document. The one comment made is that some qualitative analysis would be useful to given an intuitive perspective regarding characterizing the utility of the text-augmentation in specific settings.

== Originality == As far as the reviewers and I can tell, this is the first use of (text) retrieval-augmented chemistry related predictions. It is a relatively simple augmentation approach shown to work well in practice for two important tasks.

== Significance = First of all, the empirical results are quite promising and convincing -- validating the method and making chemistry contributions in some cases. As RAG-style systems have gained increasing popularity, it isn't unreasonable to believe that this may also be able to be integrated into general LLMs and this work code be a basis for such integration. Overall, this is a good case of applying NLP technology to other potentially impactful use cases. As the code is promised to be released, it is also lowering the barrier for others to continue work in this direction.